# Detection of AI-Created Images Using Pixel-Wise Feature Extraction and Convolutional Neural Networks

**DOI:** 10.3390/s23229037

**Published:** 2023-11-08

**Authors:** Fernando Martin-Rodriguez, Rocio Garcia-Mojon, Monica Fernandez-Barciela

**Affiliations:** AtlanTTic Research Center for Telecommunication Technologies, University of Vigo, 36310 Vigo, Spain; fmartin@uvigo.es (F.M.-R.); rocio.garcia@uvigo.es (R.G.-M.)

**Keywords:** artificial intelligence, AI images, photographs, PRNU, ELA, CCN, deep learning

## Abstract

Generative AI has gained enormous interest nowadays due to new applications like ChatGPT, DALL E, Stable Diffusion, and Deepfake. In particular, DALL E, Stable Diffusion, and others (Adobe Firefly, ImagineArt, etc.) can create images from a text prompt and are even able to create photorealistic images. Due to this fact, intense research has been performed to create new image forensics applications able to distinguish between real captured images and videos and artificial ones. Detecting forgeries made with Deepfake is one of the most researched issues. This paper is about another kind of forgery detection. The purpose of this research is to detect photorealistic AI-created images versus real photos coming from a physical camera. Id est, making a binary decision over an image, asking whether it is artificially or naturally created. Artificial images do not need to try to represent any real object, person, or place. For this purpose, techniques that perform a pixel-level feature extraction are used. The first one is Photo Response Non-Uniformity (PRNU). PRNU is a special noise due to imperfections on the camera sensor that is used for source camera identification. The underlying idea is that AI images will have a different PRNU pattern. The second one is error level analysis (ELA). This is another type of feature extraction traditionally used for detecting image editing. ELA is being used nowadays by photographers for the manual detection of AI-created images. Both kinds of features are used to train convolutional neural networks to differentiate between AI images and real photographs. Good results are obtained, achieving accuracy rates of over 95%. Both extraction methods are carefully assessed by computing precision/recall and F_1_-score measurements.

## 1. Introduction

Nowadays, generative artificial intelligence is one of the top themes of computer engineering research. The emergence of transformers [1] as a key tool for generating content has opened a world of new applications where automated systems can create productions that, until now, were exclusive to human authorship. Transformers were first used for automated translation systems, where a first processing stage (the encoder) transforms the input text into a numerical representation of text meaning; then a second stage (the decoder) converts (like an inverse transform) those intermediate data into text in another language [2].

Besides neural machine translators, other impressive applications have arisen. Famously, ChatGPT is a conversational engine created with a decoder transformer [3]. Using transformers, models for translating regular text into images have also been developed. The most known and documented examples of these last ones are DALL E [4,5] and Stable Diffusion [6]. But other examples have quickly been released, like OpenArt [7], ImagineArt [8], Adobe Firefly [9], and many others.

These artificial image generators have reached the point where they may create photorealistic images that can make humans hesitate on whether a particular image is coming from a camera or is an artificial creation. As an example, in Figure 1, three AI-created images are presented. They were created by three different engines: DALL E 2, Stable Diffusion, and OpenArt (after testing many applications, these three were found the most appropriate for photorealistic images; other models are good at producing drawings or illustrations and not so much at imitating real photographs). The prompt was the same for the three images: “realistic photo, a portrait of a dog in a library, Sigma 85 mm f/1.4”. Note that details about the lens were added (85 mm focal lens, f/1.4 numeric aperture); this is a common trick used for getting more realistic results. In the same figure, we also present three real photographs that will be processed later. The purpose of this work is to make a binary decision between two options: AI image (fully created AI image) and real image.

Another impressive AI application is Deepfake [10,11]. Deepfake can create photos and videos mixing plausible information from previous photos and/or videos. For example, creating a video of a person mixing the body of one given individual and the face of another one. The potential danger of this technology being used for fraud or other illegal purposes (defamation, pornography, etc.) has sparked much research in the field of detecting Deepfake image creation [12]. For example, in [13], Rössler et al. start by creating a large dataset of fake videos. In [14], authors exploit what is, perhaps, the most intuitive method: finding image artifacts that can reveal synthetic content. In [15], a system called “FakeCatcher” is described; this system works relying on biological signals, like the small periodic color variations present in a real face video by cause of the person’s heart rate. A very recent paper by Becattini et al. [16] presents a Deepfake detector based on Head Pose Estimation (HPE). In [17], Bappy et al. present a general image forgery detector based on recursive neural networks (LSTM). Almost all publications in this field claim that the direct use of neural networks does not produce good results in these kinds of applications. In [18], authors use the detection of “convolutional traces”, basing themselves on the fact that AI-generated images have passed several convolution stages.

The work described in this paper is similar to Deepfake detection but with a different purpose. The target is to automatically detect AI-generated photorealistic images, id est, distinguishing whole AI images from real photographs. AI images are not supposed to represent or try to represent any particular real object, place, or individual. This can be interesting for classifying images on photography websites and/or in social networks. Note that, in this case, the system is dealing with all kinds of images: human faces, animals, still nature, landscapes, etc. For this reason, it is not possible to rely on some of the “face-related” characteristics. Relying on artifacts may work for some images but not for all. Artifacts are common in artificial images within some detailed parts (the fingers of a person’s hand or pedals of a bike), but in many images, there are no visible errors. Furthermore, there are some evident errors, like a person with three hands or even with two heads, which are evident from a human view but not so easy to automate in an autonomous recognition system for any type of image.

For this particular application, there are much fewer references in the literature. In a recent preprint [19], the authors propose a method for AI image detection using a complex feature extraction based on two parallel deep learning processes. The results are similar to the ones presented in this paper, but they are using a more complex method, and their tests have been conducted on images of smaller resolution (maximum 256 × 256). In [20], authors construct a huge dataset, and they discover that systems trained on one generation model are pretty good on images from that model but not so much on others. In this work, several models are used to create the dataset, and other different models are tried for final tests (see Section 4). According to [21], features extracted for recognition are crucial in this problem when trying to work with different generation models. They point out that statistics of overexposed pixels can be a good election that seems to reinforce the election of PRNU. Another recommendation is using color-related features; the ELA pattern used in this paper is strongly related to color, as JPEG error is greater on color components. Other references [22,23,24] focus on CG (computer-generated) images, which is another type of problem as they deal with images that were created with intensive human intervention.

For a similar need, Google has recently announced a new tool called SynthID [25], which adds an invisible watermark to AI-generated images so that they can be identified. Note that this will identify AI images only if the creation engine watermarks them.

Because of the need to classify whole images with no assumption about image content, the system was designed based on methods from other image forensics applications. The main idea is to extract some relevant information from images before applying a convolutional neural network. Convolutional neural networks (CNNs) are very useful in distinguishing between classes that are visually different for humans, like digit classification [26,27], distinguishing objects relevant for making driving decisions in real traffic, and many other similar applications [28]. Nevertheless, in this case, classes are not visually different, and that suggests that the direct application of CNNs could not be very useful (besides the experience from the Deepfake case). For this reason, pixel-wise feature extraction was used. This means using processing stages that convert images into other images with the same size (it converts each pixel to a new pixel) but containing a reduced amount of information that should be relevant to the particular problem of distinguishing AI images.

There are, to date, two methods used for this issue. The first one is Photo Response Non-Uniformity (PRNU). PRNU is a kind of noise used for source camera identification (distinguishing the camera that took a given image) [29]. The origin of PRNU is the slightly different sensitivity of individual pixels in a real image sensor. This effect is due to manufacturing imperfections, and it is unavoidable. AI images should have no PRNU at all. Nevertheless, PRNU computation methods always yield a nonzero result. PRNU has been extensively studied, including its limitations [30,31]. CNN is trained to infer special characteristics of AI images with false PRNU patterns. There exist applications designed to erase or even forge PRNU patterns (embedding on an image the pattern of a given camera) [32]. So, this is a method that can reveal images created by “not very expert” or “not very malicious” users.

The second feature extraction method used is error level analysis (ELA). ELA is a special image (or pattern) that detects irregular errors in JPEG-coded images. ELA has been successfully used to detect editing in images (thus to authenticate scanned or photographed images) [33,34]. ELA has also been applied for forged face detection [35]. ELA detects non-uniformity in quantization errors due to JPEG compression. Applied to an AI-generated image, ELA normally yields a strange result, as if all pixels of the image were modified by editing. This could be due to the special nature of AI images coming from training with many JPEG-coded photographs. So, the ELA pattern is also a good choice for the application that this paper is addressing. It would seem that this method also has a limitation: all images, either coming from a real camera or an AI engine, must be obtained in the JPEG format. It would not be a great drawback as JPEG is the most frequently used photography format. Nevertheless, as seen in the remainder of this paper, ELA has been successfully tested on AI images obtained in the PNG format.

Another possible feature extraction for this problem is the local binary pattern (LBP) [36]. The LBP is based on differences between adjacent pixels. An eight-bit word is assigned to each pixel with a binary ‘1’ for greater surrounding pixels and a ‘0’ in another case. This technique has successfully been used for fake face detection [37]. The LBP is also used to detect fake face presentations to biometric systems with video replays. Patel [38] explored this approach by detecting Moiré patterns with LBP features.

The remainder of the paper is organized as follows: in Section 2, methods and processing are described, as well as the image dataset used for training and testing; in Section 3, the results are summarized. In the Section 4, the main results of this work are highlighted.

## 2. Materials and Methods

### 2.1. The Dataset

The dataset used in this work for training and testing is composed of a collection of images divided into two groups (or classes): AI-generated and real camera photographs. First, AI-generated images were created by authors using three different engines: DALL E, Stable Diffusion, and OpenArt. These images were visually checked to discard those that were not photorealistic. Second, real photos were selected randomly from image databases. There are images from the Dresden Image Database [39], from the VISION dataset [40], and also from authors’ provided images that were already used in previous studies [31,41]. There are real photos from the following cameras: Canon Ixus 70 (two instances), Casio Ex Z150 (two instances), Canon PhotoSmart SX720, Canon EOS 1100D, Kodak M1063 (two instances), and Sony ILCE 5000. Photos from smartphones are also included: Huawei P20, Huawei P9, Samsung Galaxy S3 Mini, Apple iPhone 4s, Apple iPhone 5c, Apple iPhone 6, and LG D290.

Initially, the dataset was made up of 459 AI-generated images and the same number of real photographs (a total of 918 images). Afterward, an extended dataset of 1252 was tested. Both datasets are fully balanced (the same number of samples in each class). In each test, a percentage of dataset samples will be used for training, leaving the remainder for validation.

### 2.2. PRNU Extraction

As the name, Photo Response Non-Uniformity, indicates, PRNU comes from the different light sensitivity of the different pixels (elementary sensors). This is an unavoidable characteristic due to manufacturing imperfections, and it is present on all image sensor chips. PRNU is seen as a multiplicative noise that responds to the following equation [29]:(1)$${Im}_{out}=\left(I_{ones}+{Noise}_{cam}\right).{Im}_{in}+{Noise}_{add}$$
where *Im_in_* is the “real” image presented to the camera (the incident light intensity), *I_ones_* is a matrix full of ones, *Noise_cam_* is the “sensor noise pattern” (PRNU pattern), and Im_out_ is the final image surrendered by the camera. The symbol “.” means matrix point-by-point (pixel-wise) product, and *Noise_add_* is additive noise from other sources.

PRNU is computed from an image (or from a collection of images coming from the same camera) performing a denoising process on *Im_out_*, and then computing a residual as follows:(2)$$W={Im}_{out}-denoise\left({Im}_{out}\right)$$

Neglecting the additive noise and assuming that *Im_in_ = denoise*(*Im_out_*), given a collection of images from the same camera, the PRNU pattern (sometimes called camera fingerprint) can be estimated as follows:(3)$$F=\frac{\sum_{n=1}^{N}W_{}^{n}{Im}_{in}^{n}}{\sum_{n=1}^{N}{\left({Im}_{in}^{n}\right)}^{2}}$$

Note that, in this application, we will always compute PRNU fingerprints with a single image (*N* = 1) both for AI-generated and real images. In this case, *F* = *W*/*Im_in_* (pixel-wise quotient), and it is clear that we will obtain some results, even for AI images.

Note that “denoising” is a noise reduction filter. For this problem, there are several options documented in the literature: median filter [42], Wiener filter [43], and variations of Wiener filter. In this study, a Matlab [44] implementation from [45] is used; this software uses a Wavelet Transform [46]-based Wiener filter.

From each image, a centered square 512 × 512 region is extracted to work with smaller images and to avoid logos or visible watermarks, and PRNU is computed from the sub-image. Note that the problem is classifying the whole image, not detecting a “modified” part. The minimum image size for this version is then 512 × 512. The system can be tailored easily for smaller sizes, but that would require retraining.

The results of this process are noise-like images that are very difficult to interpret visually (see Figure 2, where PRNU patterns are shown for images of Figure 1, and histogram equalization was applied to enhance these images a bit). Note that Equation (2) can be seen as a high pass filter, and so, the results contain part of image contours (a normal phenomenon when computing the pattern from a single image). There seems to be no significant visible difference between AI images, Figure 2a–c, and real ones, Figure 2d–f.

### 2.3. ELA Error Level Analysis

ELA pattern is computed to detect irregular distributions of quantization noise. This is a tool normally used to detect image editing. An ELA pattern is normally computed by coding the whole image with JPEG standard at a known, constant, and normally high-quality level (a typical value is 95%); then, the decoded image from the JPEG bit stream is subtracted from the original image.
(4)$${ELA}_{img}=img-{JPEG}^{-1}[JPEG(img,95\%)]$$

If we are facing an edited image, an irregular pattern with different intensities will appear. In Figure 3, ELA patterns for the same original images (Figure 1) are shown. Again, histogram equalization was applied to enhance these images a bit.

Note that, in this case, patterns are color images. For PRNU computation, images are converted to grayscale before any processing. Images are again cropped to the central square sub-image of size 512 × 512.

Again, a “high pass filtering” effect is evident. Visual differences between AI images, Figure 3a–c, and real ones, Figure 3d–f, are again not very remarkable. Perhaps contours are more evident in the above part, but it does not seem conclusive. Nevertheless, neural networks are able to learn differences that are not perceived by humans.

### 2.4. CNNs—Convolutional Neural Networks

CNNs are a cascade of convolutional (or linear filtering) stages accompanied by others of non-linear activation, normalization, and decimation. These stages extract high-level features from low-level data (pixels), so CNNs can process images directly with no need for feature extraction. The initial image is repeatedly filtered and decimated, creating a set of several small images that are finally processed by a classical perceptron (fully connected) stage to obtain the final result. This final result is a numerical vector of as many components as classes to be recognized. The Softmax normalization (the most frequently used at the final stage of CNNs) makes vector coefficients lie in the range of 0.0–1.0, and, in addition, they always add up to 1.0. The maximum component defines which one is the recognized class.

Filter coefficients and perceptron weights are all optimized through the training process. The training algorithm is Stochastic Gradient Descent with Momentum (SGDM) [47], which is a gradient-type optimization that minimizes the mean square error between the obtained and desired output.

In this paper, a previous image-to-image transformation is performed that acts as a pixel-wise feature extraction. This stage tries to search for relevant characteristics for distinguishing classes, removing unimportant information. As reported in the case of Deepfake detection, a direct CNN application is not good for this type of problem.

The dataset is divided randomly, selecting 85% of images of each class for training and leaving the rest for validation. Note that the dataset is balanced (it has the same number of samples for each class). A validation stage is performed at each training epoch, adequately controlling the learning process. Each complete epoch (run of all training samples in random order) is divided into *n* iterations. Each of the iterations is a mini-batch, which is a set of samples that is processed without updating weights (mini-batch size is *DatasetSize*/*n*). Testing values for *n*, optimum results were obtained for *n* = 3.

CNN structure: the number of stages and filter configuration at each stage is shown in Figure 4, and it is the same for the two kinds of pattern extraction techniques tested.

At the end of training, a confusion matrix is computed for the validation set. This means counting the number of True-Positive (AI images correctly detected), False-Negative (AI images not detected), False-Positive (real photographs detected as AI), and True-Negative (real photographs detected as real) images. The matrix is arranged in this manner:(5)$$CM=\left[\begin{array}{cc} TP & FN\\ FP & TN \end{array}\right]$$

From this matrix, several performance measurements can be computed:(6)$$Accuracy=\frac{TP+TN}{TP+TN+FP+FN}$$
(7)$$P=\frac{TP}{TP+FP}$$
(8)$$R=\frac{TP}{TP+FN}$$
(9)$$F_{1}=\frac{2\cdot P\cdot R}{P+R}$$

Accuracy is simply the success rate. The other three parameters are very easy to interpret and are very typical in classification systems: *P* (precision) would be the probability of true detection for true cases, and *R* (recall) would be the probability of effective detection of true cases. *F*_1_-score is the harmonic mean of *P* and *R*. The greater these quantities are, the better the performance achieved.

## 3. Results

CNN nets were trained and tested for both types of feature extraction. This process produces learning curves displayed in Figure 5 and Figure 6. In both cases, a good result is achieved: accuracy is 0.95 for PRNU and 0.98 for ELA. Both trainings were performed with 100 epochs. Training time is longer for the ELA case (167 min versus 109); this is reasonable because ELA images are color ones with three times more information.

Blue curves in both figures are the accuracy values obtained for each iteration (measured on the training samples), and black curves are accuracy values for the validation set at each epoch (an epoch is equal to *n* iterations, with *n* = 3 in this case). The curves below (brown and black) are the mean square error (over training and validation set); this is another method for controlling learning.

In both cases, the fact that black curves follow the evolution of blue/brown curves demonstrates that the neural network is generalizing. In the case of overfitting, the blue curve can go high, but the black curve would remain low.

Comparing both trainings, ELA offers more stable results.

The final results for both methods (confusion matrices for the validation set) are as follows:(10)$$CM(PRNU)=\left[\begin{array}{cc} 67 & 02\\ 05 & 64 \end{array}\right],\;CM(ELA)=\left[\begin{array}{cc} 66 & 01\\ 02 & 67 \end{array}\right]$$

These matrices yield the following numbers in the *P*, *R*, and *F*_1_ terms; see Table 1.

## 4. Discussion

The last presented results seem to demonstrate again that both methods are good, but ELA outperforms PRNU with a slight advantage. These results were obtained with a reduced dataset of 459 images per class. Afterward, a new test was conducted using an extended version with 626 samples per class. This test was only performed with the ELA extraction (the best option). The learning curve is presented in Figure 7. In this case, the *n* parameter was set to 5 because, with more samples, it is necessary to reduce batch size. The number of epochs is 75 because, in previous tests, it was seen that learning for ELA features was already getting stable at that point.

New assessment data for ELA features improved slightly. The confusion matrix becomes the following:(11)$$CM=\left[\begin{array}{cc} 94 & 00\\ 01 & 93 \end{array}\right]$$

The accuracy is now 0.99, precision is 0.99, recall is 1.0, and *F*1 score is 0.99. To obtain more insight into these results, we tested a pre-trained classic net. We chose AlexNet [30]. For this process, image cropping is modified so that we obtain the required size for input in this CNN: 227 × 227 × 3. The three last layers (including the final classification via a fully connected MLP layer) are modified to the new problem of binary classification (two output neurons). Weights for this level are reset to random values. The model is retrained with a very small learning rate at all levels EXCEPT at the modified ones. Learning parameters are now those recommended for this kind of training: SGDM method [48], only 4 epochs with a mini-batch size of 10 that results in 106 iterations per epoch. Again, the dataset is divided into two parts consisting of 85% for training and 15% for validation. Confusion matrixes for the validation set are the following, where the test was performed with the two feature extraction types: PRNU and ELA.
(12)$$CM(PRNU)=\left[\begin{array}{cc} 67 & 02\\ 05 & 64 \end{array}\right],\;CM(ELA)=\left[\begin{array}{cc} 90 & 04\\ 12 & 82 \end{array}\right]$$

These matrices yield the following numbers in the *P*, *R*, and *F*_1_ terms; see Table 2.

It can be seen that the method is viable but should be refined a bit. Perhaps the pre-trained levels of AlexNet are good for ordinary images but not so well fitted for PRNU/ELA patterns. Up to the point, the preferred method is ELA + specific CNN.

Another test is carried out, presenting to the original system (to the trained CNNs) a new set of completely new images. Neither was used so far in training nor validation. This new dataset consists of 150 AI-generated images and 150 other real photos. Photos were taken from unused material from the VISION database [40] (they are all smartphone photos). AI images were created using creation engines different from those of the first dataset: Leonardo.AI [49] and TensorArt [48].

Confusion matrixes for this new dataset are now as follows:(13)$$CM(PRNU)=\left[\begin{array}{cc} 128 & 22\\ 0 & 150 \end{array}\right],\;CM(ELA)=\left[\begin{array}{cc} 90 & 04\\ 0 & 150 \end{array}\right]$$

These matrices yield the following numbers in the *P*, *R*, and *F*_1_ terms; see Table 3.

Curiously, in this case, PRNU outperforms ELA. Furthermore, seeing that generally real photos are correctly classified (there are no false positives), creating a combined method is easy. If both methods are executed on the same image, it is enough that one of them classifies it as an AI image to consider it an AI image. Running the test again with this combination, the accuracy goes to 0.97 and the *F*_1_ score to 0.97. The mean execution time of this combined recognition is 0.43 s per image in the Matlab application. The implementation takes advantage of the combination “logical OR” nature: if the first method applied yields an AI image result, it is not necessary to execute the second one.

### 4.1. Conclusions

In this work, an automated system for detecting AI-created images and distinguishing them from real camera photographs was created.

Direct use of CNNs over the images seemed not very recommendable, but extracting pattern-like (or pixel-wise) features like PRNU or ELA patterns yields good results. ELA patterns work slightly better, although the combination of both methods is easy and improves results.

This issue is relatively new in the world of image forensics. Although there are many publications about the detection of image editing, including AI editing and Deepfake, pure recognition of 100% created AI images with no assumptions about content is less common. The method presented in this paper was trained with three different creation models and tested with a validation set obtained from the main dataset and also with a new dataset obtained from other different creation models. Other publications present similar results but on smaller images [19]. There also exist good results, but they are very dependent on the image creation model [20] (note that when executing a recognition, a possible creation model is not known). In [21], the authors study feature extraction methods that are able to recognize AI images coming from different models. The use of ELA and PRNU patterns is compatible with their findings.

As supplementary results,

A new dataset on AI-created images was created. This set could be augmented and published as a separate result;A graphical demo application was created; see Appendix A.

### 4.2. Future Work

Some lines of future work can be pointed out now as follows:Augmenting the AI image dataset for publication as a public research result;Enhancing that dataset by incorporating other image creation engines;Testing other pixel-wise feature extraction techniques like LBPs (local binary patterns);Testing other structures for CNN, maybe specific or pre-trained;Testing other classification schemes;Exploring the combination of methods further;Developing a version that could be used at a server to classify images uploaded to a Web 2.0 service;Trying PRNU/ELA features for Deepfake detection and other anti-forgery applications.

## Figures and Tables

**Figure 1 sensors-23-09037-f001:**
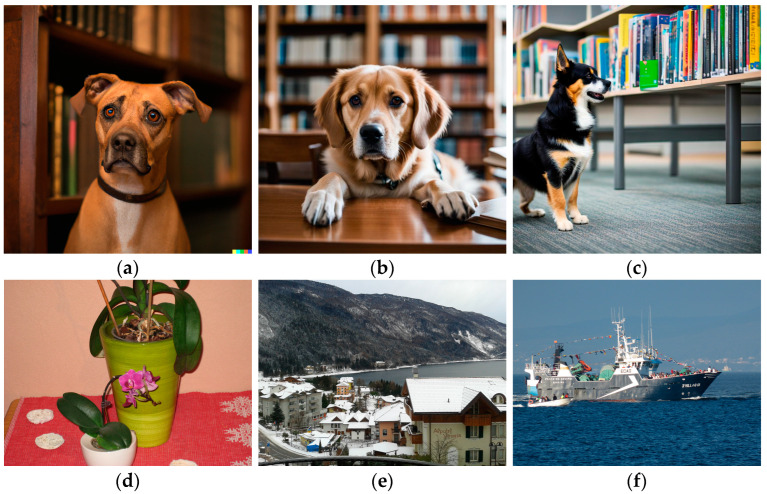
(**a**) DALL E 2 image, (**b**) Stable Diffusion image, and (**c**) OpenArt image. (**d**–**f**) Real photos.

**Figure 2 sensors-23-09037-f002:**
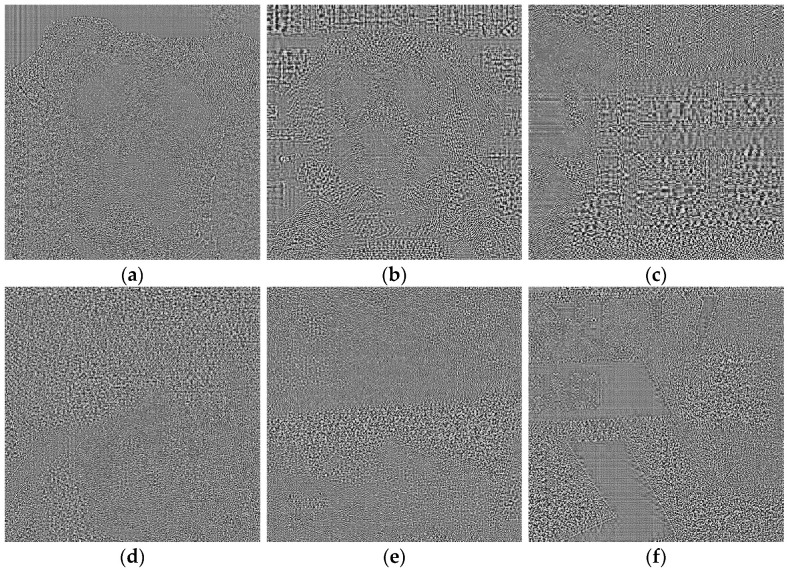
(**a**–**c**), PRNU patterns computed for AI images of Figure 1. (**d**–**f**) are examples of PRNU patterns for real images.

**Figure 3 sensors-23-09037-f003:**
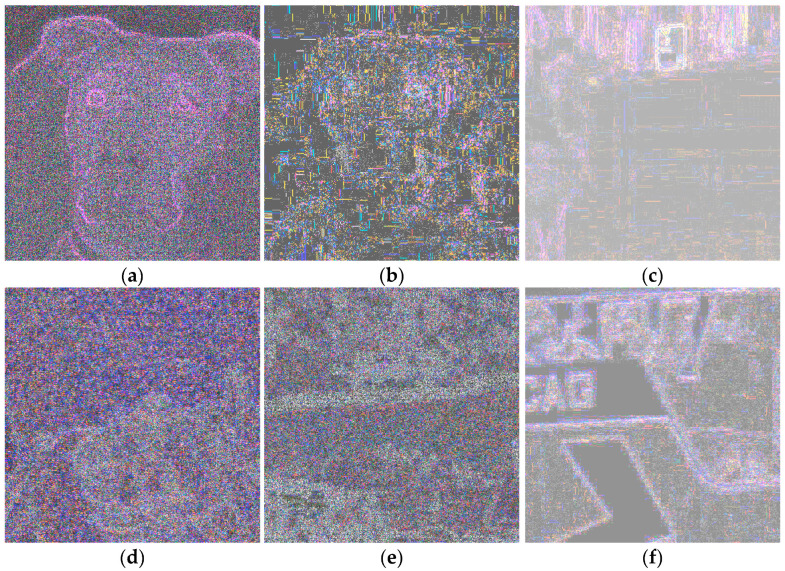
(**a**–**c**), ELA patterns computed for AI images of Figure 1. (**d**–**f**) are examples of ELA patterns for real images.

**Figure 4 sensors-23-09037-f004:**
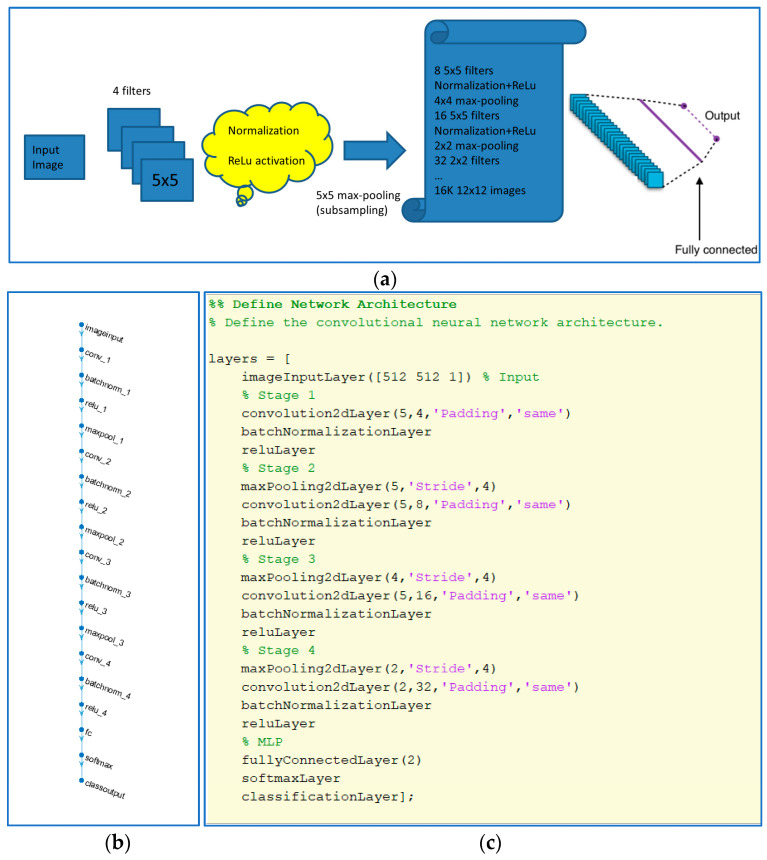
(**a**) CNN structure used. (**b**) Layers diagram. (**c**) Matlab code used to define net structure.

**Figure 5 sensors-23-09037-f005:**
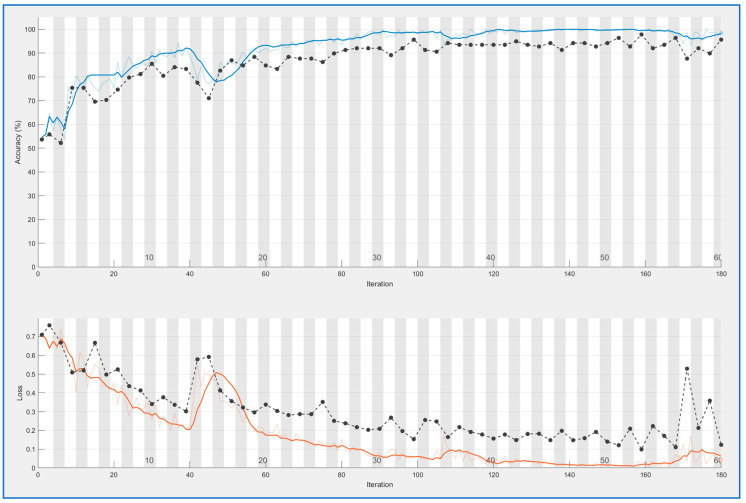
CNN training for PRNU patterns. Blue line above: accuracy for training data, black line above: accuracy for validation data. Light brown line below: mean square error for training data, black line below: mean square error for validation data.

**Figure 6 sensors-23-09037-f006:**
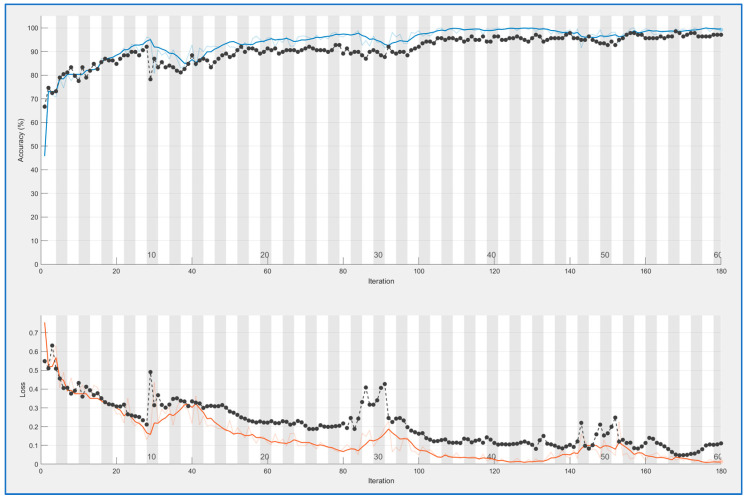
CNN training for ELA patterns. Blue line above: accuracy for training data, black line above: accuracy for validation data. Light brown line below: mean square error for training data, black line below: mean square error for validation data.

**Figure 7 sensors-23-09037-f007:**
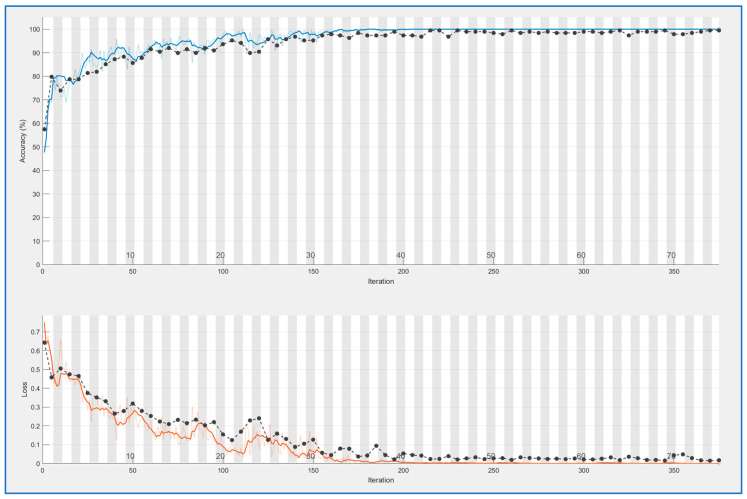
CNN training for ELA patterns (extended dataset). Blue line above: accuracy for training data, black line above: accuracy for validation data. Light brown line below: mean square error for training data, black line below: mean square error for validation data.

**Table 1 sensors-23-09037-t001:** Numerical results for both methods.

Method (Pattern Type)	Accuracy	Precision	Recall	*F*_1_ Score
PRNU	0.95	0.93	0.97	0.95
ELA	0.98	0.97	0.99	0.98

**Table 2 sensors-23-09037-t002:** Numerical results for both methods in AlexNet experiment.

Method (Pattern Type)	Accuracy	Precision	Recall	*F*_1_ Score
PRNU	0.88	0.86	0.90	0.88
ELA	0.91	0.88	0.96	0.92

**Table 3 sensors-23-09037-t003:** Numerical results for both methods (new test-only dataset).

Method (Pattern Type)	Accuracy	Precision	Recall	*F*_1_ Score
PRNU	0.93	1.00	0.85	0.92
ELA	0.91	1.00	0.82	0.90

## Data Availability

Some data are from public image databases: Dresden Image Database and Vision dataset. Other images were created by authors and are available upon request.

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
