# Peer review of "Detection of AI-Created Images Using Pixel-Wise Feature Extraction and Convolutional Neural Networks"

_sensors, 2023, doi:10.3390/s23229037_

Round 1

Reviewer 1 Report

Comments and Suggestions for Authors

The article is devoted to solving the problem of recognizing images created by AI. The topic of the article is relevant. The structure of the article does not correspond to that accepted in MDPI for research articles (Introduction (including analysis of analogues), Models and methods, Results, Discussion, Conclusions). This is the first article accepted for review where the authors did not deign to write conclusions (the last section is Discussion, which takes up a sixth of the page)! The level of English is acceptable. The article is easy to read. The figures in the article are of acceptable quality, with the exception of Fig. 2 and Fig. 3, on which it is generally impossible to get anything going. The article cites 40 sources, many of which are not relevant. The References section is designed carelessly.

The following comments and recommendations can be formulated regarding the material of the article:

1. Recognition of fake drawings traditionally occurs in the following sequence:

1) Search for alternative data streams - these could be social networks, news feeds, photographs of the same places/people/objects, etc. Here it is important to talk specifically about data streams, i.e. sets of objects that have a time stamp. This is necessary to take into account the relevance of the information being compared with the fact being verified, which, as a rule, has a later event time.

2) Context matching. Depending on the fact being verified and the nature of the data, various data matching algorithms are used in order to identify the context of a given alternative data stream with the fact being verified, i.e. to check that the alternative source of information and the person being checked are saying the same thing.

3) Verification of fact data. As soon as it is identified that the alternative and verified facts speak about the same time period, then a comparison of the immediate data values occurs.

I ask the authors to explain why they did not follow this sequence and what benefits they received from it.

2. To identify fake images, you can use the features of image quality deterioration when printing or playing on the screen. Most likely, even some local patterns will be detected in the image, albeit elusive to the eye. This can be done, for example, by calculating local binary patterns (LBP, local binary pattern) for different areas of the face after extracting it from the frame (PDF). In a nutshell, when calculating LBP, each pixel in the image and its eight neighbors are taken sequentially and their intensity is compared. If the intensity is greater than at the central pixel, it is assigned one, if less, it is assigned zero. Thus, for each pixel an 8-bit sequence is obtained. Based on the obtained sequences, a pixel-by-pixel histogram is constructed, which is fed to the input of the SVM classifier. This is a classic approach. Unlike the author's one (based on CNN), it is simple and reliable as a hammer. I ask the authors to prove the reliability of their approach.

3. In 2015, Bukinafit developed an algorithm for alternative image division into channels, in addition to the traditional RGB, for the results of which local binary patterns were again calculated, which, as in the previous method, were fed to the input of the SVN classifier. The accuracy of HTER, calculated on the CASIA and Replay-Attack datasets, was an impressive 3% at that time. At the same time, work on detecting moire appeared. Patel published a paper where he suggested looking for image artifacts in the form of a periodic pattern, caused by the superposition of two scans. The approach proved to be workable, showing an HTER of about 6% on the IDIAP, CASIA and RAFS datasets. This was also the first attempt to compare the performance of the algorithm on different data sets. And what quality indicators does the author’s system demonstrate on the mentioned datasets?

4. Unfortunately, the availability of a large number of excellent frameworks for deep learning has led to the emergence of a huge number of developers who are trying to solve the problem of identifying fake images head-on using the well-known method of ensembling neural networks. Typically this looks like a stack of feature maps at the outputs of several networks, pre-trained on some widely used dataset, which is fed to a binary classifier. In general, it is worth concluding that quite a lot of works have been published to date, which generally demonstrate good results, and which have only one small “but” in common. All these results are demonstrated within one specific dataset! The situation is aggravated by the limitations of the available data sets and, for example, on the notorious Replay-Attack, HTER 0% will not surprise anyone. All this leads to the emergence of very complex architectures, for example, like these, using various sophisticated features, auxiliary algorithms assembled in a stack, with several classifiers, the results of which are averaged, and so on... At the output, the authors get HTER = 0.04%! Unfortunately, the same “small” factor disrupts the blissful picture of the struggle for tenths of a percent. If you try to train a neural network on one data set and apply it on another, the results will turn out to be... not so optimistic. Even worse, attempts to apply classifiers in real life leave no hope at all. I ask the authors to prove that this remark does not concern their system.

Author Response

First of all, we thank very much all reviewers for their comments,
that have helped us to improve the paper significantly.

SUMMARY:
========

Purpose of the research is autmatically detect the AI generated images
distinguishing them from true camera ones. AI image means an image created with
a service like Dall-E, Stable Diffusion... All image has been created by AI from 
a text prompt. AI image does not necessarily try to fake any existing person, face, 
object or place. System has been designed as a "binary classifier", not dependant on
image content. So it does not try to dectect particular regions, result is binary:
AI-image/real-photo. That's the reason for assessing system using accuracy, precision
and recall. A segmentation system would be better assessed with measurements like
IoU (Intersection over Union).

The intended application for the system developed would be the classification of images
submitted to a server, Web 2.0 service or social network. For this problem
(binary classification of images as AI/real), there are very few references. There exists
much literacy about Deep FAke detection, where an image or a video is created with a forgery
intention using real photos of a given person.

Our system is also aimed for images of a minimum resolution: a minimum of 512 pixels in the 
smaller dimension. With these requirements, no appropriate dataset was found. For this reason,
a dataset was created, creating from scratch the AI images and obtaining real photos from
public databases (VISION, DRESDEN IMAGE DATABASE).

Full article has been review. Some English language issues have been corrected. Some
new references have been added. Discussion section has been restructured. New tests
have been done. Conclusions and future lines are integrated in the discussion part
because "oficial sensors template" recommends this if these sections are not very long,
but these subsections could be taken apart if convenient.

Outstanding additions to paper:
- Customizing a pre-trained AlexNet structure to compare it with the original CNN.
Results are not so good.
- Creating a new (smaller) dataset for testing. NO IMAGE IN COMMON WITH TRAINING DATASET.
Besides, AI images have been created using different tools. Results are only a bit worse.
What's more, this test results demonstrate that errors are generally of "false negative"
type (AI image being recognized as real photo). This suggests an easy method for integrating
the both kinds of feature extraction getting a more robust system, this possibility is
documented in the new discussion section.

SIGNIFICANT MODIFICATIONS ARE HIGHLIGTHED IN THE TEXT WITH A YELLOW BACKGROUND.
Some new figures have been added.

RESPONSE TO REVIEWER's COMMENTS:
===============================

> The article is devoted to solving the problem of recognizing images
> created by AI. The topic of the article is relevant. The structure 
> of the article does not correspond to that accepted in MDPI for research
> articles (Introduction (including analysis of analogues), Models and methods,
> Results, Discussion, Conclusions. This is the first article accepted
> for review where the authors did not deign to write conclusions
> (the last section is Discussion, which takes up a sixth of the page).

Thanksf or your comment. Discussion/conclussions have been restructured.

> The level of English is acceptable. The article is easy to read.
> The figures in the article are of acceptable quality,
> with the exception of Fig. 2 and Fig. 3, on which it is generally
> impossible to get anything going.

Thanks for your comment. PRNU/ELA patterns are merely noise images. They have been 
enhanced visually (histogram equalization) for improving a bit. Use is merely
illustrative to show that human eye is not very useful on those images.

> The article cites 40 sources,
> many of which are not relevant. The References section is designed
> carelessly.

Thanks for your comment. New reference added.

> The following comments and recommendations can be formulated
> regarding the material of the article:
> 1. Recognition of fake drawings traditionally occurs in the following
> sequence:
> 1) Search for alternative data streams - these could be social networks,
> news feeds, photographs of the same places/people/objects, etc.
> Here it is important to talk specifically about data streams,

> i.e. sets of objects that have a time stamp. This is necessary to take
> into account the relevance of the information being compared with the
> fact being verified, which, as a rule, has a later event time.

> 2) Context matching. Depending on the fact being verified and the nature
> of the data, various data matching algorithms are used in order to
> identify the context of a given alternative data stream with the fact
> being verified, i.e. to check that the alternative source of information
> and the person being checked are saying the same thing.

> 3) Verification of fact data. As soon as it is identified that the
> alternative and verified facts speak about the same time period,
> then a comparison of the immediate data values occurs.
> I ask the authors to explain why they did not follow this sequence
> and what benefits they received from it.

Thanks for your comment. AI images are understood as realistic images (or images 
that seem realistic) created fully by AI. Not necessarily with a fraud intention,
images do not necessarily represent any real person, object or place.
So a dataset had to be created, generating AI images from text... The intended use is
distinguishing these ones from real camera images. 

> 2. To identify fake images, you can use the features of image quality
> deterioration when printing or playing on the screen. Most likely,
> even some local patterns will be detected in the image, albeit elusive
> to the eye. This can be done, for example, by calculating local binary
> patterns (LBP, local binary pattern) for different areas of the face after
> extracting it from the frame (PDF). In a nutshell, when calculating LBP,
> each pixel in the image and its eight neighbors are taken sequentially
> and their intensity is compared. If the intensity is greater than at the
> central pixel, it is assigned one, if less, it is assigned zero.
> Thus, for each pixel an 8-bit sequence is obtained. Based on the obtained
> sequences, a pixel-by-pixel histogram is constructed, which is fed
> to the input of the SVM classifier. This is a classic approach.
> Unlike the author's one (based on CNN), it is simple and reliable
> as a hammer. I ask the authors to prove the reliability of their approach.

Thanks for your comment. Most times, AI images will not be printed or presented
to a camera. LBP + SVM  is used to detect editions or copy paste type modifications, 
isn't it? We are searching for a binary decision on a whole image: AI-image or real
photo. SVM is used locally. 

AI images come from transformers networks and probably will have no local artifacts
(if not edited afterwards).

Anyway, LBP is another kind of pxel-wise feature extraction that could be tested. It has
been addes to fututre lines. Thanks again for comment.

A new dataset for testing was created. WITH NO IMAGE IN COMMON WITH TRAINING DATASET.
Besides, these new AI images have been created using different tools (different than tools
used to create the training images). Results are only a bit worse. See new discussion section.

> 3. In 2015, Bukinafit developed an algorithm for alternative image
> division into channels, in addition to the traditional RGB,
> for the results of which local binary patterns were again calculated,
> which, as in the previous method, were fed to the input of the SVM
> classifier. The accuracy of HTER, calculated on the CASIA and
> Replay-Attack datasets, was an impressive 3% at that time.
> At the same time, work on detecting moire appeared. Patel published
> a paper where he suggested looking for image artifacts in the form
> of a periodic pattern, caused by the superposition of two scans.
> The approach proved to be workable, showing an HTER of about 6%
> on the IDIAP, CASIA and RAFS datasets. This was also the first attempt
> to compare the performance of the algorithm on different data sets.
> And what quality indicators does the author’s system demonstrate
> on the mentioned datasets?

Thanks for comment.
Is there any Moiré effect (aliasing) in the AI images created by Dall-E, Stable Diffussion
and other similar systems?

CASIA:
https://github.com/namtpham/casia2groundtruth/blob/master/Samples/Tp_S_NNN_S_O_pla00077_pla00077_11212.jpg
Dataset for editted images. Resolution less than 512 (minimum dimension).
It is not applicable. It is for systems that detect (and segment) editions.
If re-scaled and tested, CASIA example images result as "AI images", probably because ELA features detect
the edition. Truth table images in the database (that are binary segmentation masks) are not applicable.

RAFS-dataset..
Not found.
Found: https://paperswithcode.com/dataset/raf-db
According to homepage: "The Real-world Affective Faces Database (RAF-DB)
is a dataset for facial expression."
It is not applicable.

Replay attack:
https://paperswithcode.com/dataset/replay-attack
Database of forged faces trough replay of photos or videos in front of a camera. No AI image.
AI images need not contain a face. Nor they are replayed (or, at least it is not mandatory).
Sorry, not applicable.

IDIAP: https://www.idiap.ch/en/dataset#b_start=60&c5=all
Many datasets. Found none of AI created full images, only Deep Fake.

> 4. Unfortunately, the availability of a large number of excellent
> frameworks for deep learning has led to the emergence of a huge number
> of developers who are trying to solve the problem of identifying fake
> images head-on using the well-known method of ensembling neural networks.
> Typically this looks like a stack of feature maps at the outputs
> of several networks, pre-trained on some widely used dataset, which is fed
> to a binary classifier. In general, it is worth concluding that quite
> a lot of works have been published to date, which generally demonstrate
> good results, and which have only one small “but” in common.
> All these results are demonstrated within one specific dataset!
> The situation is aggravated by the limitations of the available
> data sets and, for example, on the notorious Replay-Attack,
> HTER 0% will not surprise anyone. All this leads to the emergence
> of very complex architectures, for example, like these,
> using various sophisticated features, auxiliary algorithms assembled
> in a stack, with several classifiers, the results of which are averaged,
>  and so on... At the output, the authors get HTER = 0.04%! Unfortunately,
> the same “small” factor disrupts the blissful picture of the struggle
> for tenths of a percent. If you try to train a neural network on one
> data set and apply it on another, the results will turn out to be...
> not so optimistic. Even worse, attempts to apply classifiers in real life
> leave no hope at all. I ask the authors to prove that this remark does not
> concern their system.

Thanks for comment. That was the reason for creating from scratch a new dataset for testing.
See results. Not documented on the paper but AI detection was tested successfully on images from
other applications: AItubo, ImagineArt (not included on study because images are not photorealistic).
Engines used in the study:
- Training/Validation: DALL E 2, Stable Diffusion, OpenArt.
- Testing: TensorArt, Leonardo.AI

Reviewer 2 Report

Comments and Suggestions for Authors

This work presents the detection of AI-created images by using two different strategies: PRNU and ELA. Although results show accuracy values over 95%, several issues have to be clarified to demonstrate applicability. The contribution of the work is not clear and seems to be limited. The obtained results seem to be good but they are not representative of such a complex task - the detection of AI-generated images.

In the literature, there are many works about AI-generated image detection, but they are not included nor analyzed in your work; thus, the advantages or novelties of your work cannot be assessed. Also, references have to be updated and increased with JCR articles.

Line 181: For clarity, mark in the original images the selected areas. How are the areas selected? In other images, how those areas are selected for an automatic classification? What is the impact of selecting different sizes of areas? What is the percentage of an AI-created image in a real image that can be detected (partial AI-created images)?    

For Figure 2, provide the real images for d), e), and f).

In section 2.4, it is said that the CNN structure is optimized. However, no evidence of the optimization process is presented. Please include the algorithm/method used, the function to be optimized, and the convergence to the optimum parameters (number and size of filters, number of layers, size of the input image, computation time, etc.).

In the discussion section, add a qualitative and quantitative comparison with similar works in order to highlight the advantages and novelties of your work.

Add and discuss the requirements and limitations of your method, for example: which is the minimum image resolution for your method? Is there a problem with images of bigger sizes?

How do you demonstrate that your results are reliable if a very small dataset (from a deep learning context) is used in the validation stage?

Can you compare your results with a pre-trained CNN (AlexNet, ResNet, GoogleNet, etc.)?

Is your method tested with AI-generated images that consider PRNU and ELA? How does your method behave under those scenarios?

The text in Figures 5 and 6 is unreadable.

Add a section for conclusions and future work. 

Comments on the Quality of English Language

Minor editing of English language required

Author Response

First of all, we thank very much all reviewers for their comments,
that have helped us to improve the paper significantly.

SUMMARY:
========

Purpose of the research is autmatically detect the AI generated images
distinguishing them from true camera ones. AI image means an image created with
a service like Dall-E, Stable Diffusion... All image has been created by AI from 
a text prompt. AI image does not necessarily try to fake any existing person, face, 
object or place. System has been designed as a "binary classifier", not dependant on
image content. So it does not try to dectect particular regions, result is binary:
AI-image/real-photo. That's the reason for assessing system using accuracy, precision
and recall. A segmentation system would be better assessed with measurements like
IoU (Intersection over Union).

The intended application for the system developed would be the classification of images
submitted to a server, Web 2.0 service or social network. For this problem
(binary classification of images as AI/real), there are very few references. There exists
much literacy about Deep FAke detection, where an image or a video is created with a forgery
intention using real photos of a given person.

Our system is also aimed for images of a minimum resolution: a minimum of 512 pixels in the 
smaller dimension. With these requirements, no appropriate dataset was found. For this reason,
a dataset was created, creating from scratch the AI images and obtaining real photos from
public databases (VISION, DRESDEN IMAGE DATABASE).

Full article has been review. Some English language issues have been corrected. Some
new references have been added. Discussion section has been restructured. New tests
have been done. Conclusions and future lines are integrated in the discussion part
because "oficial sensors template" recommends this if these sections are not very long,
but these subsections could be taken apart if convenient.

Outstanding additions to paper:
- Customizing a pre-trained AlexNet structure to compare it with the original CNN.
Results are not so good.
- Creating a new (smaller) dataset for testing. NO IMAGE IN COMMON WITH TRAINING DATASET.
Besides, AI images have been created using different tools. Results are only a bit worse.
What's more, this test results demonstrate that errors are generally of "false negative"
type (AI image being recognized as real photo). This suggests an easy method for integrating
the both kinds of feature extraction getting a more robust system, this possibility is
documented in the new discussion section.

SIGNIFICANT MODIFICATIONS ARE HIGHLIGTHED IN THE TEXT WITH A YELLOW BACKGROUND.
Some new figures have been added.

RESPONSE TO REVIEWER's COMMENTS:
===============================
> This work presents the detection of AI-created images by using two
> different strategies: PRNU and ELA. Although results show accuracy values
> over 95%, several issues have to be clarified to demonstrate applicability.
> The contribution of the work is not clear and seems to be limited.
> The obtained results seem to be good but they are not representative of
> such a complex task - the detection of AI-generated images.
> In the literature, there are many works about AI-generated image detection,
> but they are not included nor analyzed in your work; thus, the advantages
> or novelties of your work cannot be assessed. Also, references have
> to be updated and increased with JCR articles.

Thanks for your comment. New references have been added.
Nevertheless, most of the literature found is about "forgery"
detection (Deep Fake, editions...) and "pure binary decision between
AI/real-photo is much more limited.

> Line 181: For clarity, mark in the original images the selected areas.
> How are the areas selected? In other images,
> how those areas are selected for an automatic classification?
> What is the impact of selecting different sizes of areas?
> What is the percentage of an AI-created image in a real image
> that can be detected (partial AI-created images)?    

The system uses a central crop of 512x512 to perform a whole image processing.
There is no segmentation. System does not search for a forged or modified part,
it tries to know if a complete image comes from an AI generator.

> For Figure 2, provide the real images for d), e), and f).

They have been added. Thanks for your comment.

> In section 2.4, it is said that the CNN structure is optimized.
> However, no evidence of the optimization process is presented.
> Please include the algorithm/method used, the function to be optimized,
> and the convergence to the optimum parameters (number and size of filters,
> number of layers, size of the input image, computation time, etc.).

In the original version, word optimized was only used referred to CNN training:
<<Filter coefficients and perceptron weights are all optimized through the training process.>>
Anyway, some extra explanation has been added. Thanks for yur comment.

> In the discussion section, add a qualitative and quantitative comparison
> with similar works in order to highlight the advantages and novelties
> of your work.

Thanks for your comment. It is added in the conclusions subsection. Anyway, for this specific
problem, there exist less references.

> Add and discuss the requirements and limitations of your method,
> for example: which is the minimum image resolution for your method?
> Is there a problem with images of bigger sizes?

Added in section 2.2. Minimum: 512 size in the smaller dimension. Maximum: none.

>
> How do you demonstrate that your results are reliable if a very
> small dataset (from a deep learning context) is used in the validation
> stage?
>

Thanks for your comment. A new dataset has been created. Even different applications have
been used to create AI images. See results in the discussion section.
Not documented on the paper, but AI detection was also tested successfully on images from
other applications: AItubo, ImagineArt (not included on study because images are not so photorealistic).
Engines used in the study:
- Training/Validation: DALL E 2, Stable Diffusion, OpenArt.
- Final Testing: TensorArt, Leonardo.AI

> Can you compare your results with a pre-trained CNN (AlexNet,
> ResNet, GoogleNet, etc.)?

Thanks for your comment. In the discussion section a new experiment with a customized
AlexNet has been added.

> Is your method tested with AI-generated images that consider
> PRNU and ELA? How does your method behave under those scenarios?

Thanks for your comment. An expert user could try to simulate a fake PRNU, post-processing
AI images with application of reference 34. Nevertheless, this would not avoid being "catched"
by ELA features or by the combined system (described in discussion section). Is it possible to
consider PRNU and ELA in an AI image creation system? Sincerey, we don´t know... Probably,
nowadays, that kind of AI does not exist.

> The text in Figures 5 and 6 is unreadable.

Thanks for your comment. That text is printed by matlab and contains some data about training.
Those data (number of epochs, iterations per epoch..) are already on the text. Figures have been
cropped to avoid the unreadable text.

> Add a section for conclusions and future work. 

Thanks for your comment. Done.

> Comments on the Quality of English Language
> Minor editing of English language required

Thanks for your comment. Done.

Round 2

Reviewer 1 Report

Comments and Suggestions for Authors

I formulated the following comments to the previous version of the article:

1. Recognition of fake drawings traditionally occurs in the following sequence:

1) Search for alternative data streams - these could be social networks, news feeds, photographs of the same places/people/objects, etc. Here it is important to talk specifically about data streams, i.e. sets of objects that have a time stamp. This is necessary to take into account the relevance of the information being compared with the fact being verified, which, as a rule, has a later event time.

2) Context matching. Depending on the fact being verified and the nature of the data, various data matching algorithms are used in order to identify the context of a given alternative data stream with the fact being verified, i.e. to check that the alternative source of information and the person being checked are saying the same thing.

3) Verification of fact data. As soon as it is identified that the alternative and verified facts speak about the same time period, then a comparison of the immediate data values occurs.

I ask the authors to explain why they did not follow this sequence and what benefits they received from it.

2. To identify fake images, you can use the features of image quality deterioration when printing or playing on the screen. Most likely, even some local patterns will be detected in the image, albeit elusive to the eye. This can be done, for example, by calculating local binary patterns (LBP, local binary pattern) for different areas of the face after extracting it from the frame (PDF). In a nutshell, when calculating LBP, each pixel in the image and its eight neighbors are taken sequentially and their intensity is compared. If the intensity is greater than at the central pixel, it is assigned one, if less, it is assigned zero. Thus, for each pixel an 8-bit sequence is obtained. Based on the obtained sequences, a pixel-by-pixel histogram is constructed, which is fed to the input of the SVM classifier. This is a classic approach. Unlike the author's one (based on CNN), it is simple and reliable as a hammer. I ask the authors to prove the reliability of their approach.

3. In 2015, Bukinafit developed an algorithm for alternative image division into channels, in addition to the traditional RGB, for the results of which local binary patterns were again calculated, which, as in the previous method, were fed to the input of the SVN classifier. The accuracy of HTER, calculated on the CASIA and Replay-Attack datasets, was an impressive 3% at that time. At the same time, work on detecting moire appeared. Patel published a paper where he suggested looking for image artifacts in the form of a periodic pattern, caused by the superposition of two scans. The approach proved to be workable, showing an HTER of about 6% on the IDIAP, CASIA and RAFS datasets. This was also the first attempt to compare the performance of the algorithm on different data sets. And what quality indicators does the author’s system demonstrate on the mentioned datasets?

4. Unfortunately, the availability of a large number of excellent frameworks for deep learning has led to the emergence of a huge number of developers who are trying to solve the problem of identifying fake images head-on using the well-known method of ensembling neural networks. Typically this looks like a stack of feature maps at the outputs of several networks, pre-trained on some widely used dataset, which is fed to a binary classifier. In general, it is worth concluding that quite a lot of works have been published to date, which generally demonstrate good results, and which have only one small “but” in common. All these results are demonstrated within one specific dataset! The situation is aggravated by the limitations of the available data sets and, for example, on the notorious Replay-Attack, HTER 0% will not surprise anyone. All this leads to the emergence of very complex architectures, for example, like these, using various sophisticated features, auxiliary algorithms assembled in a stack, with several classifiers, the results of which are averaged, and so on... At the output, the authors get HTER = 0.04%! Unfortunately, the same “small” factor disrupts the blissful picture of the struggle for tenths of a percent. If you try to train a neural network on one data set and apply it on another, the results will turn out to be... not so optimistic. Even worse, attempts to apply classifiers in real life leave no hope at all. I ask the authors to prove that this remark does not concern their system.

The authors responded to all my comments. I found their answers quite convincing. I support the publication of the current version of the article. I wish the authors creative success.

Author Response

Thank you very much for your comments. They have helped us to improve significantly the paper.

Reviewer 2 Report

Comments and Suggestions for Authors

All the comments and suggestions have been properly addressed. This Reviewer recommends the manuscript's acceptance. 

Comments on the Quality of English Language

Minor editing of English language required

Author Response

Thank you very much for your comments. They have helped us to improve significantly the paper.

Whole language revision performed.